# Work-related stressors and mental health among LGBTQ workers: Results from a cross-sectional survey

**Benjamin Owens[1], Suzanne Mills[1]\*, Nathaniel Lewis[2], Adrian Guta[3]**

**1** School of Labour Studies, McMaster University, Hamilton, Ontario, Canada, **2** Utah Department of Health, Salt Lake City, Utah, United States of America, **3** School of Social Work, University of Windsor, Windsor, Ontario, Canada

\* smills@mcmaster.ca

## Abstract

### Purpose

Lesbian, gay, bisexual, transgender, and queer (LGBTQ) individuals experience high rates of adverse mental health outcomes due to the stressors they experience in families, communities, and society more broadly. Work and workplaces have the potential to influence these outcomes given their ability to amplify minority stress, and their ability to influence social and economic wellbeing in this already marginalized population. This study aims to identify how sociodemographic characteristics and characteristics of work, including degree of precarity, industry and perceived workplace support for LGBTQ people, influence self-reported mental health among LGBTQ people in two Canadian cities.

### Methods

Self-identified LGBTQ workers ≥16 years of age (n = 531) in Sudbury and Windsor, Ontario, Canada were given an online survey between July 6 and December 2, 2018. Multivariate ordinal logistic regression was used to calculate odds ratios (OR) to evaluate differences in gender identity, age, income, industry, social precarity, work environment, and substance use among workers who self-reported very poor, poor, or neutral mental health, compared with a referent group that self-reported good or very good mental health on a five-point Likert scale about general mental health.

### Results

LGBTQ workers with poor or neutral mental health had greater odds of: being cisgender women or trans compared with being cisgender men; being aged <35 years compared with ≥35 years; working in low-wage service sectors compared with blue collar jobs; earning < $20,000/year compared with ≥$20,000/year; working in a non-standard work situation or being unemployed compared with working in full-time permanent employment; feeling often or always unable to schedule time with friends due to work; feeling unsure or negative about their work environment; and using substances to cope with work.

addition, survey data holds sensitive information from LGBTQ workers in Windsor and Sudbury on their personal health and experiences of discrimination, including qualitative data from open-ended questions. Importantly, the LGBTQ populations in Windsor and Sudbury are relatively small, and thus publicly sharing data would compromise the level of risk of participating in the research—particularly given the marginalization that LGBTQ individuals already experience. Researchers who wish to access survey data are encouraged to contact the McMaster University Research Ethics Board to request access at ethicsoffice@mcmaster.ca, +1 (905) 525-9140 ext. 23142.

**Funding:** SM received a Partnership Development Grant from the Social Sciences and Humanities Council of Canada, Award no. 890-216-0073 (www.sshrc-crsh.gc.ca). The funders had no role in study design, data collection and analysis, decision to publish, or preparation of the manuscript.

**Competing interests:** The authors have declared that no competing interests exist.

## Conclusions

Both precarious work and unsupportive work environments contribute to poor mental health among LGBTQ people. These factors are compounded for trans workers who face poorer mental health than cis-LGBQ workers in similar environments.

## Introduction

Population-level studies have shown consistently that lesbian, gay, bisexual, transgender and other queer (LGBTQ) people have poorer mental health outcomes compared with cisgender heterosexual people [1,2]. Research has increasingly attributed these disparities to minority stress, comprising both proximal stress in the form of anticipated stigma and discrimination and distal stress in the form of actual experiences of discrimination [3,4]. Minority stress can increase depression, anxiety, deliberate self-harm, and substance abuse [1,3,4]. Although minority stress is often measured at the individual level through scales of outness, internalized homophobia and perceived discrimination [5,6], researchers have also acknowledged that experiences of minority stress may differ based on the gender and sexuality norms encountered in different spaces. These norms also have a socio-legal dimension, and thus experiences of minority stress are also dependent on structural stigma, or the level of protection and/or criminalization that individuals experience from various levels of government [3,7]. Due to the substantial amount of time spent at work, work has the potential to amplify or mitigate minority stress through both adverse experiences that LGBTQ people have in the workplace as well as work's ability to provide a safe and supportive working environment [6,8].

Previous research has shown that measures of workplace environment or climate affect the mental health outcomes of LGBTQ workers. Heterosexism in the workplace has been associated with psychological distress [9], and depression specifically [10], among gay and lesbian workers. Gay, lesbian, and bisexual workers also face sexuality-specific stressors at work, such as fear of coming out [11] or experiences of heterosexist discrimination [12,13], that can contribute to psychological distress, anxiety, and depression. Workplace harassment has also been linked with increased alcohol consumption for lesbian and bisexual women [14]. While not specifically examining workplace experiences, recent research on minority stress has also linked psychosocial stressors to poorer cardiovascular health for LGBTQ people, demonstrating the ways cumulative stress can lead to adverse physical health outcomes in addition to affecting individuals' mental health [15,16].

Alternatively, workplaces that are supportive of LGBTQ people decrease depression and anxiety [17] and job anxiety specifically [18]. Positive workplace environments, as experienced in terms of employer and co-worker support, can increase job and life satisfaction [19], as can choosing to come out at work [18]. Qualitative studies have suggested that these trends differ by sector; whereas normative gender performances in blue collar sectors (e.g., manufacturing) might negatively affect the wellbeing of sexual and gender minorities, public-sector work can be a refuge for sexual and gender minorities [8,20], though research has shown that even ostensibly progressive sectors can be sites of discrimination and stress for LGBTQ workers [21]. Furthermore, individual-level factors may also be protective; the availability of a same-sex partner and employment dyadic coping strategies appears to attenuate the relationship between workplace stress and anxiety [22].

In addition to the above factors, mental health outcomes among LGBTQ people might also be affected by unemployment and precarious work in ways similar to that of cis-heterosexual

workers workers [23]. While previous studies have acknowledged that economic insecurity affects the mental health of bisexual people similarly to that of heterosexual and cisgender individuals [24], they do not necessarily acknowledge that adverse outcomes at the population level could be exacerbated by the overrepresentation of LGBTQ people in low-wage service work [25] or unemployment resulting from discrimination in hiring and firing practices [26].

A large volume of research has explored the effects of precarious employment, work that is insecure and/or unstable, on worker wellbeing [27]. Population level studies have associated precarious employment with poor mental health [28–30]. Rodgers [31] identified four key dimensions of precarious employment including short, limited working arrangements (instability), low control over wages or work conditions (job insecurity), lack of institutional protection, and low income/poverty. Some scholars have added additional dimensions including, degree control over work, status or prestige, risk of exposure to physical hazards, training and career advancement opportunities, and socio-cultural environment at work [27]. Though employment contingency or deviations from the standard employment relationship are commonly used to measure precarity in quantitative studies, studies are increasingly using composite measures and subjective measures [32–34]. In the case of LGBTQ workers, socio-cultural environments that are supportive could feasibly constitute a dimension of precarity. For conceptual clarity in this paper, however, precarity is understood as job instability resulting from contingent work arrangments and job insecurity resulting from the absence of control over wages or work hours.

Studies adopting a variety of objective, subjective and composite measures have shown that workers in precarious employment relationships have greater likelihood of suicidal ideation, depression, poor sleep and psychological distress [23,32,33]. Precarious employment and social stress leads to adverse psychological and physiological health through several direct and indirect pathways. Precarious employment negatively affects health both directly by increasing the release of stress hormones and indirectly by decreasing life and job satisfaction, causing material deprivation, fostering substance abuse, or increasing exposure to work hazards [27]. The relationship between mental health and precarity, moreover, exists on a continuum, with more precarious jobs leading to poorer mental health compared with less precarious jobs [35].

Intersectionality can also affect how one experiences precarity. Social location such as racialization, immigration status, gender identity and sexual orientation has the potential to mediate the effects of precarity on health. In a population level study in South Korea gender mediated the relationship between non-standard and chronic disease conditions; non-standard employment was associated with mental disorders in women and musculoskeletal disorders and liver disease in men [36]. Second, marginalized groups, such as women, recent immigrants and LGBTQ people, are often over-represented in precarious work and as a result more likely to suffer negative health effects. To date, however, most studies examining gender and precarity have adopted binary conceptions of gender and overlooking transgender, genderqueer, and non-binary population [36]. Precarious employment may amplify minority stress experienced by trans workers, in particular, since it is associated with fewer job protections, less access to extended health benefits, and greater economic insecurity. Sexual minority workers who are unemployed, for example, are more likely to report mental illness [37]. Although the link between unemployment and poor mental health has also been reported in research where sexual orientation is not reported [38], the reasons for which LGBTQ workers become unemployed (e.g., discrimination, work-related stress) may be different from heterosexual workers. Factors such as low income may also reinforce disadvantage. In Canada, bisexual people situated below the low-income cut-off in Canada were more likely to experience depression and to perceive discrimination due to both early-in-life experiences (e.g., discrimination) that affected financial stability, and difficulties accessing supports and mental health care due to their financial position [24].

Among studies exploring the relationship between work and LGBTQ mental health, few have been designed specifically to assess mental health outcomes across diverse sociodemographic segments of the LGBTQ community, diverse types of work, or to measure the relative impacts of precarious work, unemployment, *and* workplace culture on LGBTQ mental health outcomes.

Hypothesis: LGBTQ people in precarious work and in work environments that are not supportive of LGBTQ identities are more likely to have poor mental health outcomes.

By locating itself in two cities with transitional economies, Sudbury and Windsor [39,40] this study also seeks to better understand the specific challenges faced by LGBTQ workers in post-industrial societies marked by increasing precarity and still-persistent discrimination.

## Methods

### Ethics

This study was approved by the McMaster University Research Ethics Board (MREB#1866). For participants who completed the survey online, the informed consent process preceded the survey and participants provided their consent through answering an online question that asked them whether they consent to participate in the research. Participants who completed paper surveys provided written consent prior to beginning the survey. Consent from parents or guardians of participants ages 16 and 17 was not required given the potential social risks of involving family members in research about LGBTQ identity, and this was cleared by the McMaster University Research Ethics Board.

### Recruitment and sample

We collected survey responses for 662 individuals in Sudbury and Windsor from July 6 to December 2, 2018. Surveys were available in both English and French in web-based and paper formats. We used rigorous multi-faceted recruitment strategies, including respondent-driven sampling techniques to reduce sampling bias, since LGBTQ people constitute a 'hidden population' [41]. Paper surveys were distributed by community organizations in each city. Links to the web-based survey were distributed using a variety of strategies including e-mailing membership lists of several large unions and employment lists in each city, distributing postcards to LGBTQ support groups promoting the survey on local radio stations, and placing ads on Facebook, Instagram, and the geosocial meet-up platforms Grindr and Scruff. Community advisory committees were formed in each city and members distributed surveys through their personal networks.

These strategies were supplemented by in-person recruitment at Pride and other community events in both cities, where researchers were available with tablets as well as paper surveys to encourage participation and minimize selection bias. Principles of respondent driven sampling were used to reach networks outside the reach of the above methods: participants were encouraged to distribute the survey to members of their social network by unique codes and a prize incentive [42]. Eligible survey participants had worked in one of the two cities in the past year, were ≥16 years of age, and identified as LGBTQ. Eligibility criteria was determined to be met based on a brief pre-survey, which asked participants for their age, whether they self-identified as LGBTQ, and whether they had worked in Windsor or Sudbury over the past year. Given the exploratory nature of the study, which focused on the work and community experiences of LGBTQ workers, non-LGBTQ persons were excluded from participating.

### Variables

Based on the literature on LGBTQ mental health and workplace mental health, we used ordered logistic regression to model self-reported mental health using sociodemographic

characteristics (gender identity, age and income), industry, indicators of precarity (employment relationship, difficulty scheduling time with friends, fear of job loss), perceived workplace support for LGBTQ workers (workplace environment), and substance use, alcohol use and tobacco use as additional risk factors.

**Outcome variable.** Mental health: A self-assessed, single-item mental health outcome variable was determined based on responses to a question asking participants to rate their mental health on a five-point Likert scale. This scale is based on other single-item measures of self-rated mental health commonly used in health surveys as it reduces respondent burden and has been shown to be a reliable measure of mental health [43]. Poor and very poor were recoded to a single 'poor' category; average was maintained as 'neutral'; and good and very good were recoded into a single referent 'good' category.

**Predictor variables: Sociodemographic characteristics.** Gender identity: Participants identified their gender identity from a list, with the option to write-in their own response. This variable was recoded into three categories: cisgender men, cisgender women, and trans. Trans women, trans men and non-binary/genderqueer identities were collapsed into one category 'trans' to increase cell sample size because of the small number of responses in each individual category.

Age range: Participants wrote in their year of birth, which was then recoded into two age groups: <35 years and ≥35 years.

Income: Participants were asked to indicate how much money they made in the previous year by choosing from a range of categories in $10,000 increments, before taxes and deductions; responses were recoded into two categories: < $20,000/year and ≥$20,000/year.

Race/ethnicity: Participants were asked to self-identify the race and/or ethnicity that best described them from a list, with the option to write-in their own response. This variable was then coded into four categories: White, Black, Indigenous, and other racialized. Respondents who did not identify as White, Black, or Indigenous were combined into one category 'other racialized' due to the small number of responses in each individual category.

**Predictor variables: Work characteristics.** Two variables were used to measure precarious work: a subjective measure (social life affected by uncertain work schedule), and an objective measure (employment relationship). Industry and work environment were used to capture the industries participants worked in and the degree to which their workplaces were positive for LGBTQ workers.

Social life affected by uncertain work schedule: Participants selected how often uncertainty about their work schedules limited time with friends, family, or community activities from a five-point scale; rarely, never, and sometimes were recoded into a single referent category; often and always were recoded into a single category.

Employment relationship: Participants selected what best matched their employment relationship in their primary job from a list of options. This was recoded into three categories: full-time permanent, non-standard (comprising contract, temporary, self-employed, and part-time work), and unemployed.

Industry: Participants selected the industry they worked in for their primary job based on the North American Industry Classification System (NAICS). Mining, manufacturing, transportation, agriculture, and construction were recoded into a single category of blue collar; finance, administration, information, management, real estate, education, health, and public administration were recoded as white collar; food service, retail, arts/entertainment, and other service were recoded as low-wage service.

Work environment: Participants were asked to rate their work environment for LGBTQ workers from a five-point scale; very negative, negative, and unsure were recoded to negative/unsure, and positive and very positive were recoded to positive.

**Predictor variables: Other risk factors.** Substance use: Participants were asked if they have ever used substances to cope with their work. This was coded as a binary variable.

Alcohol use: Participants were asked if they have ever used alcohol to cope with their work. This was coded as a binary variable.

## Statistical analyses

A multivariable ordered logistic regression model was used to estimate the association between mental health and individual characteristics. We compared workers who reported poor mental health, neutral mental health and good mental health, which served as the referent category. Participants with complete outcome and descriptive data were included in regression models, and respondents with missing values were excluded from analysis. We created a multivariable regression model using a purposeful selection strategy [44], first conducting exploratory univariate analyses to assess whether there was a relationship between each variable and mental health outcomes. Variables meeting $p < 0.05$ in univariate analyses were retained in the multivariable model. Results of a partial likelihood ratio test comparing the full model (including alcohol use) with the more parsimonious one suggests that adding the alcohol use variable did not result in significantly improved model fit (LR chi2(4) = 0.35 Prob >chi2 = 0.5542). We conducted a variance inflation factor (VIF) test for multicollinearity and no variables exceeded a VIF of 3, indicating that collinearity is minor and that no variables merited further investigation. Results are presented as adjusted odds ratios (aOR) and associated 95% confidence intervals (CIs). STATA™ (College Station, TX: StataCorp, LLC) was used to perform the analysis.

## Results

Of the 632 people who completed the survey (405 from Sudbury and 266 from Windsor), data from 531 individuals had no missing values and was available for analysis. Overall, 48.2% reported good mental health, 32.8% reported neutral mental health, and 19.0% reported poor mental health.

## Univariate analysis

Demographic, socioeconomic and employment characteristics among those who reported poor, neutral and good mental health, the referent group, are reported in Table 1. Compared with the referent group, those who reported poorer mental health were more likely to be cisgender women (OR: 2.15; 95% CI: 1.47–3.14) or trans (OR: 4.29; 95% CI: 2.68–6.85), and less likely to be cisgender men. Those reporting poor or neutral mental health were also more likely to be aged <35 years (OR: 3.58; 95% CI: 2.44–5.27) compared with people aged ≥35 years, and to have higher odds of having an individual income of ≤$20,000/year (OR: 5.05; 95% CI: 3.52–7.27), compared with >$20,000/year. Respondents reporting poorer mental health did not differ significantly from the referent group based on race/ethnicity.

Compared with the referent group, those reporting poorer mental health also had higher odds of working in the low-wage service sector (OR: 3.14; 95% CI: 1.77–5.57), compared with working in a blue collar sector; odds of working in a white collar sector (OR: 1.36; 95% CI: .79–2.35) were not significantly different. Those reporting poorer mental health also had higher odds of being unemployed (OR: 25.22; 95% CI: 5.2–122.29) or being in a non-standard work situation (OR: 2; 95% CI: 1.44–2.78), compared with permanent, full-time employment.

Compared with the referent group, those reporting poorer mental health had higher odds of rating their work environment for LGBTQ employees negatively or being unsure what to rate it (OR: 3.16; 95% CI: 2.19–4.54), compared with rating it positively. Compared with the referent group, they also had higher odds of often or always feeling unable to schedule time with friends

**Table 1. Sociodemographic, economic, and employment characteristics of LGBTQ workers by reported mental health: Good (ref., n = 256), neutral (n = 174), and poor (n = 101), and univariate ordered logistic regression models for each variable ('neutral' and 'poor' mental health compared to 'good' referent category).**

| | Self-reported mental health | | | | |
| | Good | Neutral | Poor | | |
| Characteristics | N(%) | N(%) | N(%) | OR | 95% CI |
|---|---|---|---|---|---|
| Gender identity | | | | | |
| Cis man (ref) | 112 (43.8) | 51 (29.3) | 15 (14.9) | ref | |
| Cis woman | 111 (43.4) | 86 (49.4) | 49 (48.5) | 2.15* | 1.47–3.14 |
| Trans | 33 (12.9) | 37 (21.3) | 37 (36.6) | 4.29* | 2.68–6.85 |
| Age range | | | | | |
| ≥35 years (ref) | 109 (42.6) | 39 (22.4) | 10 (9.9) | ref | |
| <35 years | 147 (57.4) | 135 (77.6%) | 91 (90.1) | 3.58* | 2.44–5.27 |
| Race/ethnicity | | | | | |
| White | 193 (75.4) | 135 (77.6) | 76 (76.8) | ref | |
| Black | 20 (7.8) | 8 (4.6) | 7 (7.1) | 0.76 | 0.39–1.51 |
| Indigenous | 28 (10.9) | 26 (14.9) | 13 (13.1) | 1.19 | 0.74–1.92 |
| Other racialized | 15 (5.9) | 5 (2.9) | 3 (3) | 0.51 | 0.21–1.21 |
| Industry | | | | | |
| Blue collar (ref) | 40 (15.6) | 15 (8.6) | 9 (8.9) | ref | |
| White collar | 153 (59.8) | 101 (58.1) | 37 (36.6) | 1.36 | 0.79–2.35 |
| Low-wage service | 63 (24.6) | 58 (33.3) | 55 (54.6) | 3.14* | 1.77–5.57 |
| Income | | | | | |
| Over $20,000/year (ref) | 217 (84.8) | 105 (60.3) | 38 (37.6) | ref | |
| Under $20,000/year | 39 (15.2) | 69 (39.7) | 63 (62.4) | 5.05* | 3.52–7.27 |
| Employment relationship | | | | | |
| Permanent full-time (ref) | 151 (59) | 75 (43.1) | 35 (34.7) | ref | |
| Non-standard | 105 (41) | 97 (55.8) | 59 (58.4) | 2.00* | 1.44–2.78 |
| Unemployed | 0 (0) | 2 (1.2) | 7 (6.9) | 25.22* | 5.20–122.29 |
| Workplace environment | | | | | |
| Positive (ref) | 212 (82.8) | 121 (69.5) | 50 (49.5) | ref | |
| Unsure/negative | 44 (17.2) | 53 (30.5) | 51 (50.5) | 3.16* | 2.19–4.54 |
| Social life affected by work | | | | | |
| Never/rarely/sometimes (ref) | 213 (83.2) | 131 (75.3) | 55 (54.5) | ref | |
| Often/always | 43 (16.8) | 43 (24.7) | 46 (45.5) | 2.74* | 1.88–3.99 |
| Substance use to cope with work | | | | | |
| Does not use (ref) | 131 (51.2) | 79 (45.4) | 35 (34.7) | ref | |
| Uses substances | 125 (48.8) | 95 (54.6) | 66 (65.4) | 1.56* | 1.13–2.16 |
| Alcohol use to cope with work | | | | | |
| Does not use (ref) | 196 (49.4) | 134 (33.8) | 67 (16.9) | ref | |
| Uses alcohol | 60 (44.9) | 40 (29.9) | 34 (25.4) | 1.34 | 0.93–1.94 |

because of work (OR: 2.74; 95% CI: 1.88–3.99) compared with sometimes, rarely, or never. Finally, compared with the referent group, those reporting poorer mental health had higher odds of using substances to cope with work (OR: 1.56; 95% CI: 1.13–2.16). Respondents reporting poorer mental health did not differ significantly from the referent group in using alcohol to cope with work.

## Multivariable analysis

All variables for which significant differences in odds were observed in multivariate analysis were retained in the multivariable model (Table 2). An additional industry category (white

**Table 2. Ordered logistic regression estimates of impact of sociodemographic and employment characteristics on LGBTQ workers' mental health ('neutral' and 'poor' mental health compared to 'good' referent category).**

| Characteristics | aOR | 95% CI |
|---|---|---|
| Gender identity | | |
| Cis man (ref) | ref | |
| Cis woman | 1.91* | 1.27–2.89 |
| Trans | 3.01* | 1.82–4.97 |
| Age range | | |
| ≥35 years (ref) | ref | |
| <35 years | 2.08* | 1.32–3.26 |
| Industry | | |
| Blue collar (ref) | ref | |
| White collar | 1.67 | 0.92–3.02 |
| Low-wage service | 1.98* | 1.05–3.72 |
| Income | | |
| Over $20,000/year (ref) | ref | |
| Under $20,000/year | 2.85* | 1.86–4.37 |
| Employment relationship | | |
| Permanent full-time (ref) | ref | |
| Non-standard | 1.07 | 0.73–1.56 |
| Unemployed | 9.45* | 1.64–54.64 |
| Workplace Environment | | |
| Positive (ref) | ref | |
| Unsure/negative | 2.25* | 1.51–3.33 |
| Social life affected by work | | |
| Never/rarely/sometimes (ref) | ref | |
| Often/always | 2.00* | 1.33–3.00 |
| Substance use to cope with work | | |
| Does not use (ref) | ref | |
| Uses substances | 1.48* | 1.04–2.11 |

collar) for which differences from the referent (blue collar) were not observed was also retained as it was likely to underpin differences in variables such as employment status, income, and work environment. In the multivariable analysis, those reporting poorer mental health had greater adjusted odds of being trans (aOR: 3.01; 95% CI: 1.82–4.97) or cisgender women (OR: 1.91; 95% CI: 1.27–2.89), compared with cisgender men. Compared with the referent group, those reporting poorer mental health were more likely to be aged <35 years (OR: 2.08; 95% CI: 1.32–3.26) compared with people aged ≥35 years, and to have higher odds of having an individual income of ≤$20,000/year (OR: 2.85; 95% CI: 1.86–4.37), compared with >$20,000/year.

Compared with the referent group, those reporting poorer mental health continued to have higher odds of working in a low-wage service job (aOR: 1.98; 95% CI: 1.05–3.72), compared with working in a blue collar position; odds of working in a white collar job (aOR: 1.67; 95% CI: .92–3.02) were higher than observed in the univariate analysis but still not significant. Those reporting poorer mental health also had higher odds of being unemployed (aOR: 9.45; 95% CI: 1.64–54.62) compared with permanent, full-time employment; odds of being in a non-standard work situation were not significant.

Compared with the referent group, those reporting poorer mental health had higher odds of rating their work environment negatively or being unsure what to rate it (aOR: 2.25; 95% CI: 1.51–3.33). In addition, those reporting poor mental health had higher odds of often or

always feeling unable to schedule time with friends because of work (OR: 2.00; 95% CI: 1.33–3.00) compared with sometimes, rarely, or never. Finally, compared with the referent group, those reporting poorer mental health had higher odds of using substances to cope with work (aOR: 1.48; 95% CI: 1.04–2.11).

## Discussion

Our analysis shows that LGBTQ workers reporting poorer mental health are more likely to be cisgender women or trans people, to be <35 years old, to work in low-wage service sector jobs, and to have incomes under $20,000/year. They are also more likely to rate their work environment as negative for LGBTQ employees or to be unsure about it, and to work in precarious work environments, as evidenced by higher odds of working in non-standard employment relationships and often or always being unable to schedule time with friends due to work scheduling. LGBTQ workers reporting poor mental health are also more likely to lack sufficient coping strategies, as they are more likely to use substances to cope with work.

The narrative in LGBTQ organizational studies has long been that blue collar industries will be the most negative for workers' mental health; our study shows that it is in fact low-wage, customer-facing workers who have poorer mental health. This suggests that the challenges experienced by LGBTQ in work environments that are gendered as masculine may be partially offset by the greater stability, income security and benefits that are often associated with these jobs. It also suggests that low-wage service work, while having less rigid gender norms and an overrepresentation of LGBTQ workers, also involves elements of precarity, low income, and customer interaction that can ultimately be detrimental to the mental health of LGBTQ people. This finding is important because LGBTQ people, due to histories of discrimination in the relatively secure industrial sectors, are overrepresented in the low-wage service jobs that now characterize post-industrial, economically transitioning areas.

Relatedly, our results extend organizational research about LGBTQ inclusion that has often focused on the importance of supportive work cultures to the exclusion of other aspects of work quality. Our results show that supportive work cultures are protective measures for LGBTQ mental health, however, other work characteristics, are also important. Precarity, which is endemic to the locales in this study, emerged as a clear driver of poor mental health among LGBTQ people, as it is for the population as a whole. Those employed on a temporary, part-time or casual basis, or whose uncertain work schedules limit their times with friends, may have greater income fluctuation, less job security, and fewer employment related benefits causing poor mental health.

While precarious work affects LGBTQ people in ways that are similar to cisgender heterosexual population, it may be experienced differently or more severely by different segments of the LGBTQ community, particularly trans workers and to a lesser degree lesbian and bisexual women, who both had higher odds of poor mental health compared with cisgender gay and bisexual men. In the case of trans workers, the greater economic security, stability and extended health benefits typically associated with full-time permanent employment might help mediate distal or proximal stressors experienced at work or outside of work by providing workers with the ability to access mental health supports or take leaves. Additionally, if workers feel insecure at work and fear negative repercussions such as being assigned fewer shifts, they may be less likely to be open about their sexual or gender identity or to report discrimination with negative consequences for their mental health. In addition, people with lower incomes in precarious employment are likely to experience greater economic insecurity and be at greater risk of poor mental health than those with higher incomes. This finding

corroborates and extends previous research showing that health effects of employment precarity are amplified for low-income populations [45] and women [33].

Our findings also challenge the idea of younger LGBTQ generations being more empowered; rather, mental health problems are on the rise among young people generally and for the LGBTQ workers in this study specifically. The higher frequency of poor mental health for this group may be brought on by economic insecurity and by working in low-wage service work environments where there is less investment in worker wellbeing compared with other sectors. This finding reaffirms previous research on LGBTQ youth and mental health, which has consistently shown relatively poor mental health outcomes compared to heterosexual and cisgender youth, and potentially greater risk of poor mental health among younger LGBTQ people compared with LGBTQ adults [46–48].

Although we did not include sexual orientation as a variable (i.e., separating lesbian, gay, bisexual, and pansexual identities), preliminary analysis confirmed previous research showing that bisexual people had more negative mental health outcomes than gay and lesbian respondents. Despite these findings, sexual orientation was omitted because it was conceptually confounded with gender identity and, therefore, was excluded to maintain a parsimonious model. Indeed, the terminology for sexual orientation used in the survey was not able to capture differences in sexual orientation among non-binary transgender respondents, since the labels 'lesbian' and 'gay' presume that one fits the gender binary, and were reported as pansexual. Additionally, data on race and ethnicity was omitted from the multivariate analysis since univariate analysis was insignificant.

## Conclusions and limitations

Previous attention to LGBTQ mental health and work from a policy perspective has often focused on ensuring that workplaces are LGBTQ friendly. The results of this study reaffirm the importance of LGBTQ supportive workplaces for mental health while also calling attention to the importance of income and job security. Both inclusion and quality of work more generally are captured in the International Labour Organization's concept of decent work, defined as "productive work for women and men in conditions of freedom, equity, security and human dignity" [49]. Remedies to poor mental health among LGBTQ populations, particularly trans people and low-income LGBTQ people, therefore, need to extend beyond employer and union driven inclusion strategies to address both the concentration of LGBTQ people in precarious work and the degradation of work more broadly. While targeted programs and strategies can address the former, the latter requires more widespread institutional change. Regulatory environments that require the provision of paid sick days, discourage employers from irregular scheduling and subcontracting, and facilitate collective organizing would have positively affect LGBTQ workers' mental health. Additionally, state funded security measures such as easily accessible unemployment insurance, state pensions and universal mental health care (including mental health) would also benefit low-wage LGBTQ workers most at risk of poor mental health.

In the current Canadian context, results suggest that LGBTQ mental health supports should target unemployed and precariously employed LGBTQ people, alongside trans and low-income individuals. This includes community-level mental health supports that are free or low-cost for LGBTQ people who do not have access to mental health care through employment, and that accommodate irregular scheduling to ensure low-wage service workers have access. Employment training also needs to be integrated with mental health supports to provide pathways for LGBTQ youth to move into more stable, well-paid employment.

This study has some limitations. LGBTQ people with poor mental health might be more or less likely to be unemployed or employed in non-standard employment, as such there is the potential of a reverse relationship between mental health and labour market outcomes. While we sought to reduce recruitment bias by using in person, as well as digital recruitment techniques, the use of an internet survey and social media plateforms as a form of recruitment may have introduced selection bias in favour of younger, digitally literate respondents. We used a single-item scale for mental health which, though considered a reliable measure that decreases the burden for survey participants, lacks the complexity of multi-item measures [43]. Our study also did not fully capture the effect of education on mental health outcomes because of suspected collinearity with income and industry, nor were we able to measure the effects of sexual orientation, which was omitted due to aforementioned conceptual limitations in the survey. Additionally, in our analysis of the data we used complete case analysis and excluded respondents with missing values, which may have introduced bias into the model. Last, results may not be generalizable to the United States or to larger cities with greater levels of acceptance of LGBTQ people. We suspect that results are, however, generalizable since growth in services and specifically the low-wage service sector has been widespread across the Global North.

## Supporting information

**S1 File.**
(DOCX)

## Acknowledgments

Adriane Paavo, United Steelworkers; United Steelworkers local 6500; Sarah McCue, Unifor; Mélodie Bérubé and Scott Florence, Sudbury Workers Education and Advocacy Centre; and Paul Chislet, Windsor Workers Education Centre provided in-kind support and assistance with data collection; Randy Jackson provided guidance with survey design and community engagement. Laur O'Gorman; John Antoniw; Bobby Jay Aubin; Derrick Carl Biso; Dani Bobb; Vincent Bolt; Lynne Descary; Debra Dumouchelle; Mel Jobin; Paul Pasanen; Jennifer Johnson; Leah McGrath-Reynolds; Angela Di Nello; and Natalie Oswin assisted with instrument design and data collection. Niko Yiannakoulias provided guidance with statistical analysis and revisions.

## Author Contributions

**Conceptualization:** Suzanne Mills, Nathaniel Lewis.

**Formal analysis:** Benjamin Owens.

**Funding acquisition:** Suzanne Mills.

**Project administration:** Suzanne Mills.

**Visualization:** Benjamin Owens.

**Writing – original draft:** Suzanne Mills, Nathaniel Lewis.

**Writing – review & editing:** Benjamin Owens, Suzanne Mills, Nathaniel Lewis, Adrian Guta.

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
