## [Decision Letter · Decision Letter 0]

19 Jul 2022

PONE-D-22-12783Work-related stressors and mental health among LGBTQ workers: Results from a cross-sectional surveyPLOS ONE

Dear Dr. Mills,

Thank you for submitting your manuscript to PLOS ONE. After careful consideration, we feel that it has merit but does not fully meet PLOS ONE’s publication criteria as it currently stands. Therefore, we invite you to submit a revised version of the manuscript that addresses the points raised during the review process.

We look forward to receiving your revised manuscript.

Kind regards,

Remya Lathabhavan

Academic Editor

PLOS ONE

Journal Requirements:

Additional Editor Comments :

Thanks for considering PLOS ONE. Based on reviews received, a major revision is suggested .

Reviewers' comments:

Reviewer's Responses to Questions

**Comments to the Author**

1. Is the manuscript technically sound, and do the data support the conclusions?

Reviewer #1: Yes

Reviewer #2: Yes

2. Has the statistical analysis been performed appropriately and rigorously? 

Reviewer #1: Yes

Reviewer #2: Yes

3. Have the authors made all data underlying the findings in their manuscript fully available?

Reviewer #1: No

Reviewer #2: No

4. Is the manuscript presented in an intelligible fashion and written in standard English?

Reviewer #1: Yes

Reviewer #2: Yes

5. Review Comments to the Author

Reviewer #1: This is a well written manuscript, that I have enjoyed reading. The researchers are Weill organized and well informed. I specifically enjoyed the methodology and analysis parts. The discussion part is well written and reflective of the study expected outcomes.

Reviewer #2: Thank you for the opportunity to review “Work-related stressors and mental health among LGBTQ workers: Results from a cross-sectional survey.”

In general, it’s more appropriate in the context of discussing stressors and their effects to describe populations as “marginalized” rather than “vulnerable.” Marginalized is what happens to a group, vulnerable more often describes an innate quality.

Why wasn’t sexual orientation appropriately measured and included? The literature often shows that persons with non-monosexual identities (e.g., bisexual, pansexual, etc) fair worse than persons with monosexual identities (e.g., straight, gay, lesbian).

Why was race excluded from the analysis? The assumption that including it would introduce bias is an incomplete assumption that requires further support if its to pass muster.

Abstract:

Purpose: consider replacing “vulnerable” with “marginalized” when describing LGBTQ populations.

Methods: Include how and when respondents were surveyed. Describe referent group in methods. What constitutes “good mental health” in the context of this study?

Results: Delete “compared with a referent group with good mental health” as this doesn’t not clarify the results subsequently presented.

Conclusion: Clear and concise.

Introduction:

Page 3, Line 60: Please use a more recent reference to describe LGBTQ well-being; 2008 is already outdated by more than a decade.

Page 3, Line 61: you’ve reversed the definition of distal (external) and proximal (internal) stressors. Please verify your definitions at the outset.

Page 3, Line 65: turn to Meyer and Hatzenbuehler to describe additional distal stressors that occur on a more global scale (e.g., policies affecting LGBTQ life): https://pubmed.ncbi.nlm.nih.gov/?term=hatzenbuehler+stigma

Page 3, Line 70: I would note that there are disparities in access to employment for LGBTQ persons. UPDATE: I see this is well discussed later in the introduction at Line 93.

Page 4, Line 84: there are also differential experiences across sectors, even among health professions: https://pubmed.ncbi.nlm.nih.gov/28537796/

Page 5, Line 95: great introduction to factors associated with wellbeing related to work environment.

Page 5, Line 116: I suggest taking Minority Stress Theory further to include physical health outcomes as they relate to stress. This has been explored more recently in cardiovascular health as it relates to minority stress and disparities in risk factors as a result of stress:

https://www.ahajournals.org/doi/10.1161/CIR.0000000000000914

https://www.ahajournals.org/doi/full/10.1161/CIR.0000000000001003

Page 6, Line 112: desase = disease

Methods:

Great recruitment methods to reach LGBTQ persons.

How was LGBTQ identity ascertained?

Why were non-LGBTQ persons excluded?

Page 8, Line 173: Modeled how?

Page 8, Line 179: their mental what?

Page 9, Line 191: why was 35 the dividing point?

Page 9, Line 195: Why is income divided at $20,000? Why not a poverty level to account for variation between cities?

Page 10, Line 217: work environment is way too subjective here. Why not include measures of existing workplace protections? Explicit support of LGBTQ persons, etc?

Page 10, Line 218: why wasn’t tobacco considered as a substance for coping with stress?

Page 10, Line 227: style suggestion, “participants with complete outcome and descriptive data were included in regression models.” Excluding missing data may in fact introduce bias; Line 228-230 is not an appropriate assumption.

Results:

Page 11, Line 249: style suggestion, “those with poorer mental health” is actually “those who reported poorer mental health.” There are differences by gender in how people report mental health. As no objective measure of mental health were utilized, I caution saying who had poor mental health and instead frame it as those who reported it. People can have poor mental health without reporting it.

Otherwise, a clear presentation of results.

Discussion:

Page 16, Line 316: this study doesn’t have a variable for “male-dominated industries” so cannot speak to whether or not this is a factor in the mental health of the respondents.

Page 18, Line 360: why was the measure for sexual orientation not able to accurately capture respondent’s sexual identity? What preliminary analyses were conducted? Please ensure this is included in supplementary materials.

Page 18, Line 367: the exclusion of race data is highly problematic and flattens the experience of persons with multiply marginalized identities.

Conclusions and Limitations.

These suggestions to improve the work environment are lackluster. Be more specific and discuss institutional as well as policy-level interventions.

6. PLOS authors have the option to publish the peer review history of their article (what does this mean?). If published, this will include your full peer review and any attached files.

Reviewer #1: No

Reviewer #2: No

---

## [Author Response · Author response to Decision Letter 0]

5 Sep 2022

We are very grateful to the editor and reviewers for your thoughtful comments on our manuscript. We address each of the comments in greater detail below. If there are any other comments or points that arise during the review process, please let us know.

Editor’s comments:

a. The manuscript has been reviewed to ensure that it meets the PLOS ONE style requirements, including naming conventions for files.

a. Thank you for this comment. Parental consent was not required for participants ages 16 and 17 given the potential social risks of involving family members in research about LGBTQ+ identity. This was cleared by the McMaster Research Ethics Board. The manuscript’s methods section has been updated to include this information. 

3. In your Data Availability statement, you have not specified where the minimal data set underlying the results described in your manuscript can be found. PLOS defines a study's minimal data set as the underlying data used to reach the conclusions drawn in the manuscript and any additional data required to replicate the reported study findings in their entirety. All PLOS journals require that the minimal data set be made fully available. Important: If there are ethical or legal restrictions to sharing your data publicly, please explain these restrictions in detail. Please see our guidelines for more information on what we consider unacceptable restrictions to publicly sharing data: http://journals.plos.org/plosone/s/data-availability#loc-unacceptable-data-access-restrictions. Note that it is not acceptable for the authors to be the sole named individuals responsible for ensuring data access. 

a. The Data Availability statement has been updated to describe, in detail, the ethical restrictions that limit our ability to share the data publicly. Researchers who are looking to access the underlying data are instructed to contact the McMaster Research Ethics Board and are provided with the relevant contact information. The revised statement reads as follows:

“Data cannot be shared publicly as participants did not consent to the release of survey data in a public domain, nor was the public sharing of data cleared by the McMaster University Research Ethics Board. In addition, survey data holds sensitive information from LGBTQ workers in Windsor and Sudbury on their personal health and experiences of discrimination, including qualitative data from open-ended questions. Importantly, the LGBTQ populations in Windsor and Sudbury are relatively small, and thus publicly sharing data would compromise the level of risk of participating in the research—particularly given the marginalization that LGBTQ individuals already experience. Researchers who wish to access survey data are encouraged to contact the McMaster University Research Ethics Board to request access at ethicsoffice@mcmaster.ca, +1 (905) 525-9140 ext. 23142.”

a. The manuscript’s methods section has been updated to include an ethics subsection, which stipulates that the project was cleared by the McMaster Research Ethics Board (MREB#1866), and the ways consent was obtained for the online and paper surveys.

Reviewer #2’s comments:

1. In general, it’s more appropriate in the context of discussing stressors and their effects to describe populations as “marginalized” rather than “vulnerable.” Marginalized is what happens to a group, vulnerable more often describes an innate quality.

a. Thank you for this comment. We have replaced the word ‘vulnerable’ with ‘marginalized’ in the manuscript.

2. Why wasn’t sexual orientation appropriately measured and included? The literature often shows that persons with non-monosexual identities (e.g., bisexual, pansexual, etc) fair worse than persons with monosexual identities (e.g., straight, gay, lesbian). 

a. Thank you for drawing attention to this important topic. There are two reasons why sexual orientation was excluded from our analysis. First, given our small sample size, we needed to make strategic decisions on what to include in order to balance building a comprehensive model while also ensuring the model was appropriately parsimonious. This was further complicated by our need to include workplace factors, which were our primary variables of interest. Secondly, while we are aware that bisexual people typically experience poorer mental health outcomes when compared to heterosexual, gay, and lesbian individuals, we also experienced a conceptual challenge that bolstered our decision to not include the sexual orientation variable; since the labels ‘lesbian’ and ‘gay’ presume that one fits the gender binary, people with a non-binary gender identity become classified as pansexual, regardless of differences in their attraction. Therefore, the variables of sexual orientation and gender identity are not conceptually distinct. Based on these two reasons, we excluded sexual orientation from our modelling, which is consistent with some other research on self-reported mental health and LGBTQ+ individuals (Streed Jr. et al., 2018). The manuscript has been updated to add greater detail to this reasoning. We have also expanded the limitations section of the manuscript to include this as a limitation of the study.

3. Why was race excluded from the analysis? The assumption that including it would introduce bias is an incomplete assumption that requires further support if it’s to pass muster. 

a. Thank you for this very important comment. To address this, we have included the race and ethnicity variable in the univariate analysis. Due to a lack of statistical significance, the variable was omitted from the final model. We have removed from the manuscript any references to reporting bias in race/ethnicity data.

4. Abstract: Purpose: consider replacing “vulnerable” with “marginalized” when describing LGBTQ populations.

a. As stated above, all uses of the word ‘vulnerable’ to describe a group have been replaced with ‘marginalized’.

5. Abstract: Methods: Include how and when respondents were surveyed. Describe referent group in methods. What constitutes “good mental health” in the context of this study?

a. The methods section of the abstract has been updated to state that the survey was administered online between July 6 and December 2, 2018, and that ‘good mental health’ was a self-rated measure on a five-point Likert scale that asked about participants’ mental health in general.

6. Abstract: Results: Delete “compared with a referent group with good mental health” as this doesn’t not clarify the results subsequently presented

a. This sentence has been removed from the results section of the abstract.

7. Page 3, Line 60: Please use a more recent reference to describe LGBTQ well-being; 2008 is already outdated by more than a decade.

a. The citation from 2008 has been replaced with a more recent article on LGBTQ+ well-being.

8. Page 3, Line 61: you’ve reversed the definition of distal (external) and proximal (internal) stressors. Please verify your definitions at the outset.

a. Thank you for catching this. This line has been revised to reflect the proper definitions of distal and proximal stressors. 

9. Page 3, Line 65: turn to Meyer and Hatzenbuehler to describe additional distal stressors that occur on a more global scale (e.g., policies affecting LGBTQ life): https://pubmed.ncbi.nlm.nih.gov/?term=hatzenbuehler+stigma

a. We appreciate this suggestion and have added a sentence on structural stigma and the socio-legal dimensions of minority stress.

10. Page 4, Line 84: there are also differential experiences across sectors, even among health professions: https://pubmed.ncbi.nlm.nih.gov/28537796/

a. This is important nuance. We have added a sentence on how, even in sectors that are ostensibly progressive, LGBTQ workers can experiences stressors. 

11. Page 5, Line 116: I suggest taking Minority Stress Theory further to include physical health outcomes as they relate to stress. This has been explored more recently in cardiovascular health as it relates to minority stress and disparities in risk factors as a result of stress:

https://www.ahajournals.org/doi/10.1161/CIR.0000000000000914
https://www.ahajournals.org/doi/full/10.1161/CIR.0000000000001003

a. Thank you for this suggestion. The physiological effects of minority stress are now included in the literature review.

12. Page 6, Line 112: desase = disease 

a. This spelling mistake has been corrected. 

13. Great recruitment methods to reach LGBTQ persons. How was LGBTQ identity ascertained? 

a. The manuscript has been updated to explain that, prior to completing the survey, participants were required to complete a pre-survey to determine eligibility. This survey included a question on whether they self-identified as LGBTQ.

14. Why were non-LGBTQ persons excluded?

a. The methods section has been updated to explain that, given the exploratory nature of the study which focused on the work and community experiences of LGBTQ workers, non-LGBTQ persons were excluded from participating. Indeed, the experiences of non-LGBTQ people were outside of the scope of this study. 

15. Page 8, Line 173: Modeled how?

a. This line has been updated to explain that the data was modelled using ordered logistic regression. 

16. Page 8, Line 179: their mental what?

a. Thank you for catching this. This line has been updated to specify mental ‘health’. 

17. Page 9, Line 191: why was 35 the dividing point? 

a. We chose 35 as the dividing point for the age variable since previous research on career trajectories has identified 35 as the beginning of ‘midlife’ (Ferraro et al., 2018). In addition, many unions’ young workers’ committees define ‘young workers’ as those under the age of 35.

18. Page 9, Line 195: Why is income divided at $20,000? Why not a poverty level to account for variation between cities?

a. The survey question for income was ordinal, and asked participants to select the income range that best reflected their income. The options for this question were in $10,000 increments—following Statistics Canada—and, given the lack of granularity in the measure, the $20,000 cut-off best represented participants living below the poverty line in both cities.

19. Page 10, Line 217: work environment is way too subjective here. Why not include measures of existing workplace protections? Explicit support of LGBTQ persons, etc.?

a. Thank you for this comment. While we understand your concern, we believe that the subjective measure of work environment (which encompasses participants’ perceptions of policy, attitudes of co-workers and management, and symbolic supports) is appropriate here for two reasons. First, polices and supports vary across workplaces and industries, and given the project’s varied industrial makeup, a self-reported measure allows for a single variable that accounts for participants’ perceptions of their work environment across sectors. In addition, there is evidence that workplace protections can be superficial and do not always make workers feel safe, particularly given the primacy of LGBTQ workers’ interpersonal relationships with co-workers, customers, and management in determining comfort at work (Tayar, 2017; Giuffre et al., 2008). Thus, a subjective measure of workplace environment allowed participants to take a holistic approach when considering their levels of safety and comfort at work. 

20. Page 10, Line 218: why wasn’t tobacco considered as a substance for coping with stress?

a. The variable used to measure substance use included whether or not participants had used tobacco to cope with work, alongside other substances. Separate variables measuring the use of individual substances were excluded to maintain a parsimonious model. 

21. Page 10, Line 227: style suggestion, “participants with complete outcome and descriptive data were included in regression models.” 

a. Thank you for this style suggestion. We have incorporated this proposed change into the manuscript.

22. Excluding missing data may in fact introduce bias; Line 228-230 is not an appropriate assumption.

a. Thank you for this very important point. We have removed the claim that complete case analysis removes bias, and have added this as a limitation to the study at the end of the manuscript. 

23. Page 11, Line 249: style suggestion, “those with poorer mental health” is actually “those who reported poorer mental health.” There are differences by gender in how people report mental health. As no objective measure of mental health were utilized, I caution saying who had poor mental health and instead frame it as those who reported it. People can have poor mental health without reporting it.

a. Thank you for this important comment. The manuscript has been updated to keep results consistent with the self-reported nature of the mental health variable.

24. Page 16, Line 316: this study doesn’t have a variable for “male-dominated industries” so cannot speak to whether or not this is a factor in the mental health of the respondents.

a. The phrase ‘male-dominated industries’ has been replaced with the language used in the industry variable, i.e., ‘blue collar industries’. 

25. Page 18, Line 360: why was the measure for sexual orientation not able to accurately capture respondent’s sexual identity? What preliminary analyses were conducted? Please ensure this is included in supplementary materials. 

a. See response #2 above.

26. Page 18, Line 367: the exclusion of race data is highly problematic and flattens the experience of persons with multiply marginalized identities. 

a. See response #3 above. We have included race data in the univariate results, though this variable did not meet the conditions for inclusion in the final model.

27. Conclusions and limitations: These suggestions to improve the work environment are lackluster. Be more specific and discuss institutional as well as policy-level interventions. 

a. Thank you for this comment. We have expanded the conclusion to include more specific interventions.

References:

Ferraro, H. S., Prussia, G., & Mehrotra, S. (2018). The impact of age norms on career transition intentions. Career Development International, 23(2), 212-229.

Giuffre, P., Dellinger, K., & Williams, C. L. (2008). “No retribution for being gay?”: Inequality in gay-friendly workplaces. Sociological Spectrum, 28(3), 254-277.

Streed Jr, C. G., McCarthy, E. P., & Haas, J. S. (2018). Self-reported physical and mental health of gender nonconforming transgender adults in the United States. LGBT health, 5(7), 443-448.

Tayar, M. (2017). Ranking LGBT inclusion: Diversity ranking systems as institutional archetypes. Canadian Journal of Administrative Sciences/Revue Canadienne des Sciences de l'Administration, 34(2), 198-210.

---

## [Decision Letter · Decision Letter 1]

26 Sep 2022

Work-related stressors and mental health among LGBTQ workers: Results from a cross-sectional survey

PONE-D-22-12783R1

Dear Dr. Mills,

We’re pleased to inform you that your manuscript has been judged scientifically suitable for publication and will be formally accepted for publication once it meets all outstanding technical requirements.

Kind regards,

Remya Lathabhavan

Academic Editor

PLOS ONE

Additional Editor Comments (optional):

Reviewers' comments:

Reviewer's Responses to Questions

**Comments to the Author**

1. If the authors have adequately addressed your comments raised in a previous round of review and you feel that this manuscript is now acceptable for publication, you may indicate that here to bypass the “Comments to the Author” section, enter your conflict of interest statement in the “Confidential to Editor” section, and submit your "Accept" recommendation.

Reviewer #2: All comments have been addressed

Reviewer #3: All comments have been addressed

2. Is the manuscript technically sound, and do the data support the conclusions?

Reviewer #2: Yes

Reviewer #3: Yes

3. Has the statistical analysis been performed appropriately and rigorously? 

Reviewer #2: Yes

Reviewer #3: Yes

4. Have the authors made all data underlying the findings in their manuscript fully available?

Reviewer #2: No

Reviewer #3: Yes

5. Is the manuscript presented in an intelligible fashion and written in standard English?

Reviewer #2: Yes

Reviewer #3: Yes

6. Review Comments to the Author

Reviewer #2: (No Response)

Reviewer #3: (No Response)

7. PLOS authors have the option to publish the peer review history of their article (what does this mean?). If published, this will include your full peer review and any attached files.

Reviewer #2: No

Reviewer #3: No

---

## [Editor Report · Acceptance letter]

17 Oct 2022

PONE-D-22-12783R1 

Work-related stressors and mental health among LGBTQ workers: Results from a cross-sectional survey 

Dear Dr. Mills:

I'm pleased to inform you that your manuscript has been deemed suitable for publication in PLOS ONE. Congratulations! Your manuscript is now with our production department. 

Kind regards, 

on behalf of

Dr. Remya Lathabhavan 

Academic Editor

PLOS ONE